# Vav proteins do not influence dengue virus replication but are associated with induction of phospho-ERK, IL-6, and viperin mRNA following DENV infection *in vitro*

Evangeline Cowell,[1] Hawraa Jaber,[1] Luke P. Kris,[1] Madeleine G. Fitzgerald,[1] Valeria M. Sanders,[1] Aidan J. Norbury,[2] Nicholas S. Eyre,[1] Jillian M. Carr[1]

**ABSTRACT** Vav proteins are guanine exchange factors that activate Rac1/RhoA signaling to regulate many biological processes including inflammatory responses. Here, the expression and functional significance of Vav's were investigated during dengue virus (DENV) infection. In primary monocyte-derived macrophages, Vav1, −2, and −3 mRNA levels demonstrate variable responses to DENV infection and correlate with DENV-induced host inflammatory (IL-6 and TNF-α), antiviral (viperin), or cell adhesion [intercellular cell adhesion molecule (ICAM-1)] mRNA induction. Strong positive correlations were seen in particular for Vav2 with TNF-α and Vav3 with IL-6 mRNA. In the retinal pigmented epithelial cell line, ARPE-19, Vav2 was the main Vav expressed and was not affected by DENV infection. Heterologous Vav1 expression in ARPE-19 cells induced an increase in basal IL-6 mRNA but did not enhance DENV-induced mRNA responses. DENV RNA and DENV-induced viperin and IL-6 mRNA responses were also unaffected by Vav2 siRNA knockdown. Treatment of DENV-infected ARPE-19 cells with EHop-016 to block Vav signaling did not affect DENV RNA levels but increased DENV-mediated induction of IL-6 mRNA. More detailed assessment of DENV-induced responses to azathioprine, a clinically used immunosuppressant that can also block Vav signaling and act as a nucleoside analog, similarly demonstrated no change in DENV RNA levels but resulted in inhibition of DENV-induced phospho-ERK and increased DENV-induced-IL-6 and viperin mRNA in ARPE-19 cells. Thus, levels of Vav are associated with DENV-induced inflammatory responses, and blocking Vav signaling pathways does not compromise control of viral replication but may influence DENV-induced host responses.

**IMPORTANCE** Dengue disease is characterized by an inflammatory-mediated immunopathology, with elevated levels of circulating factors including TNF-α and IL-6. If the damaging inflammatory pathways could be blocked without loss of antiviral responses or exacerbating viral replication, then this would be of potential therapeutic benefit. The study here has investigated the Vav guanine exchange factors as a potential alternative signaling pathway that may drive dengue virus (DENV)-induced inflammatory responses, with a focus on Vav1 and 2. While Vav proteins were positively associated with mRNA for inflammatory cytokines, blocking Vav signaling didn't affect DENV replication but prevented DENV-induction of p-ERK and enhanced IL-6 (inflammatory) and viperin (antiviral) mRNA. These initial data suggest that Vav proteins could be a target that does not compromise control of viral replication and should be investigated further for broader impact on host inflammatory responses, in settings such as antibody-dependent enhancement of infection and in different cell types.

**KEYWORDS** dengue virus, Vav, inflammation, azathioprine, interleukin-6, viperin

Address correspondence to Jillian M. Carr, jill.carr@flinders.edu.au.

The authors declare no conflict of interest.

See the funding table on p. 15.

Vav proteins (Vav1, 2, and 3) are a family of highly homologous proteins that are guanine exchange factors (GEF) and exchange GDP for GTP on Rac1 and RhoA proteins (1). This activates Rac1/RhoA which stimulates a variety of signaling pathways including activation of NFkB (2) and signal transducers and activators of transcription (STAT)−3 pathways (1, 3, 4). Vav-associated Rac1/RhoA activation can influence biological processes including inflammation, cell growth, and migration (1, 5). Vav1 is primarily expressed in hematopoietic cells (6) and can drive B- and T-cell development and immune responses (7), Toll-like receptor 4 (TLR4) and lipopolysaccharide responses on B-cells and macrophages (8, 9), FcϒR-mediated phagocytosis in macrophages (10), and changes in macrophage morphology (11). Vav2 and 3 are more ubiquitously expressed and associated with diverse functions such as maintaining the integrity of the microvasculature (12), blood pressure control, glucose homeostasis, and neurodevelopment (13–15).

The pathogenesis of dengue disease is associated with a vascular leak syndrome, where increased proinflammatory factors released from cells such as dengue virus (DENV)-infected macrophages act upon endothelial cells to reduce vascular integrity (16, 17). Both of these facets of DENV pathogenesis are relevant to the above roles of Vav as mediators of inflammation and maintenance of vascular integrity in these key cell types. Further Vav proteins are involved in cellular processes that are known to be part of the host response to DENV-infection, such as TLR4 signaling which is stimulated by DENV NS1 protein (18–20), and JNK and STAT3 that are activated during DENV infection and can mediate DENV induction of cytokines and chemokines (21). There is evidence that Vav proteins can affect cell signaling during other viral infections, through interactions with viral proteins such as HIV nef (22, 23), murine hepatitis virus (MHV)−68 M2 (24), and Epstein Barr virus nuclear antigen 1 (EBNA1) protein (25, 26), and influencing the viral replication cycle for instance by induction of MHV-68 latency in B-cells (27) and increased HTLV-I proviral load (28–30). Additionally, Vav is important in T-cell activation and induction of HIV transcription (31) and is phosphorylated by HTLV-I proteins to induce a leukemic state. Increased Vav levels and Vav phosphorylation are also associated with Vav2 binding of the L2 protein of bovine papilloma virus and have been proposed to affect viral entry and oncogenic potential (32). In relation to DENV, in a screen to identify cellular receptors for DENV, Vav1 was identified as a virus-binding partner in mosquito cells and proposed to be involved intracellularly in facilitating DENV entry (33). In the study here, the expression of Vav mRNA and proteins following DENV infection in monocyte-derived macrophages (MDMs), a key target for DENV replication *in vivo*, and ARPE-19 cells, which are relevant to DENV-mediated ocular inflammation, have been defined. Responses were analyzed that reflect inflammatory (TNF-α and IL-6) and antiviral (viperin) processes. The effect of manipulating levels of Vav protein or treatment with drugs to target Vav-Rac1 signaling pathways on these responses was defined. Outcomes demonstrate that Vav proteins are not antiviral, but inhibiting Vav can reduce DENV-induction of the signaling mediator phospho-ERK (p-ERK) and enhance DENV-induction of IL-6 and viperin mRNA. This justifies further scrutiny of these pathways as potential influencers of the DENV-induced pathogenic inflammatory responses.

## MATERIALS AND METHODS

### Cells

ARPE-19 human retinal pigment epithelial cell line was cultured in Dulbecco modified Eagle medium (DMEM):F12 media supplemented with 10% (vol/vol) fetal calf serum (FCS, Thermo Fisher Scientific-Gibco), penicillin (50 units/mL), streptomycin (50 µg/mL), and 1% (vol/vol) GlutaMAX (Thermo Fisher Scientific-Gibco) at 37°C in 5% $CO_2$. To generate primary macrophages, cells were isolated from anonymous healthy volunteer donors from buffy coats provided by the Australian Red Cross Blood Service with approval from the Southern Adelaide Clinical Human Research Ethics Committee (SACHREC,

approval number, HREC/19/SAC/199). Cells were collected by density centrifugation (Lymphoprep), and monocytes were isolated using EasySep Human Monocyte Isolation Kit (Stemcell) by immunomagnetic negative selection. Isolated cells were cultured in DMEM with 10% (vol/vol) FCS, penicillin (50 units/mL), streptomycin (50 µg/mL), 1% (vol/vol) GlutaMAX, and 5% HS (Heat inactivated human sera, isolated from donors, SACHREC approval number HREC/16/SAC/306) and differentiated into adherent MDMs over 5 days.

## Virus stocks

DENV infections were performed using DENV-2 Mon601 and generated from a full-length, infectious clone from the mouse-brain adapted New Guinea C isolate (34). Viral RNA was *in vitro* transcribed, transfected into baby hamster kidney (BHK-21) fibroblasts and amplified in C6/36 mosquito cells. Viral stocks were clarified by centrifugation, filtered, and stored at $-80°C$ for use, with titers determined by plaque assay on Vero cells as described previously (35).

## Infection with DENV

ARPE-19 or MDMs were seeded in six-well plates the day prior to infection and challenged with DENV at a multiplicity of infection (MOI) = 1 for ARPE-19 or MOI = 3 for MDMs, in serum-free medium. Cells were incubated with the inoculum for 90 min at 37°C in 5% $CO_2$ with rocking of the plates every 15 min. After incubation, the inoculum was removed, and cell monolayers washed once with phosphate-buffered saline (PBS, Gibco) then cultured in complete media. For experiments with drug treatment, cells were pre-treated for 2 h with 50 µM azathioprine (Selleck Chemicals) or 1 µM EHop-016 (Selleck Chemicals) prior to infection as above, with subsequent culture of the cells in the presence of drug. Uninfected or no drug treatment (vehicle, 0.01% dimethyl sulfoxide [DMSO]) controls were performed in parallel, and samples were collected at the indicated time point post-infection (pi) for analysis.

## Transfection of retinal cells

ARPE-19 cells were seeded the day prior to transfection with pC.HAVav1 (a gift from Joan Brugge, Addgene plasmid #14553) or control plasmids using Lipofectamine 3000 (Invitrogen) as per manufacturer's instructions for 4 h. Cells were allowed to recover in complete media for 1 h then challenged with DENV (MOI = 0.1) at 37°C for 3 h. The infection inoculum was then removed and replaced with complete media, and samples were collected at the indicated time point pi for analysis. For Vav2 siRNA treatment, cells were reverse transfected with on-Target plus human Vav2 siRNA or non-targeting control siRNA (NTC), SMARTPool (Dharmacon) using Dharmafect 1 transfection reagent, allowed to recover then challenged with DENV, as above.

## RNA extraction, reverse transcription, and qRT-PCR

Cells were lysed, and total RNA was extracted using TRIzol Reagent (Thermo Fisher Scientific-Ambion) according to the manufacturer's instructions. Extracted RNA was DNaseI-treated, and the concentration and purity were measured using a Nanodrop 2,000 spectrophotometer (Thermo Fisher Scientific). 500 ng of RNA was reverse transcribed using Random Primer Mix (New England Biolabs) and M-MuLV reverse transcriptase (New England Biolabs). qRT-PCR was performed on the resultant cDNA using PowerUp SYBR Green Master Mix (Thermo Fisher Scientfic) with 1 µM of each primer (Table 1). Real time qRT-PCR was performed with a hot-start protocol using a Rotor-Gene real-time PCR system (Qigaen) with the following settings: 1 cycle of 95°C for 5 min, 40 cycles of 95°C for 15 s, 58°C for 30 s, and 72°C for 30 s; and 1 cycle of 72°C for 60 s followed by melt curve analysis. Results were normalized against a housekeeping gene (cyclophilin) and relative mRNA abundance determined by the ΔCt method.

**TABLE 1** Primer sequences used for qRT-PCR

| mRNA target (human) | Primer pair | Product size (bp) |
|---|---|---|
| DENV-2 | Forward: 5′ -GCAGATCTCTGATGAATAACCAAC- 3′ Reverse: 5′ -TTGTCAGCTGTTGTACAGTCG- 3′ | 102 |
| Cyclophilin | Forward: 5′ -GGCAAATGCTGGACCCAACACAAA- 3′ Reverse: 5′ -CTAGGCATGGGAGGGAACAAGGAA- 3′ | 355 |
| Viperin | Forward: 5′ -CCAGAGCAGGAACAAGGG- 3′ Reverse: 5′ -GTGAGCAATGGAAGCCTGATC- 3′ | 84 |
| TNF-α | Forward: 5′ -CCCCAGGGACCTCTCTCTAATC- 3′ Reverse: 5′ -GGTTTGCTACAACATGGGCTACA- 3′ | 98 |
| IL-6 | Forward: 5′ -AGACAGCCACTCACCTCTTCAG- 3′ Reverse: 5′ -TTCTGCCAGTGCCTCTTTGCTG- 3′ | 132 |
| Factor B | Forward: 5′ -ACTGAGCCAAGCAGACAAGC- 3′ Reverse: 5′ -AGAAGCCAGAAGGACACACG- 3′ | 288 |
| ICAM-1 | Forward: 5′ -TAAGCCAAGAGGAAGGAGCA- 3′ Reverse: 5′ -CATATCATCAAGGGTTGGGG- 3′ | 280 |
| Vav 1 | Forward: 5′ -TCAGTGCGTGAACGAGGTCAAG- 3′ Reverse: 3′ -CCATAGTGAGCCAGAGACTGGT- 3′ | 102 |
| Vav 2 | Forward: 5′ -CTGCTGTTCCACAAGATGACCG- 3′ Reverse: 5′ -AGCCCTGCTTTCCTTGAAGGTG- 3′ | 115 |
| Vav 3 | Forward: 5′ -GAGTGGAGTCAGCCATCTCTAG- 3′ Reverse: 5′ -CACGTTGCATAGGAACCACAAGC- 3′ | 141 |

## Immunostaining and fluorescence microscopy

Cells were grown on 0.2% gelatine-coated glass coverslips, infected as above, and, at 48 h post-infection (hpi), fixed in 1% (vol/vol) formalin. Fixed cells were permeabilized [0.05% (vol/vol) IGEPAL (Sigma)], blocked [5% human sera, 4% normal goat sera, and 0.4% bovine serum albumin in PBS (vol/vol)], and immunostained with anti-DENV NS1 (4G4, mouse monoclonal antibody hybridoma supernatant, 2 µg/mL, 1/10 dilution) and either total Vav (rabbit anti-Vav, 59 ng/mL, 1/2000 dilution, Abcam, ab40875) or Vav2 (rabbit polyclonal anti-Vav2, 20 µg/mL, 1/50 dilution, InVitrogen, SC68-03). Bound antibody was detected using donkey anti-mouse-Alexa488 or goat anti-rabbit-Alexa555 antibodies, and the nuclei were stained with Hoechst 33342 (5 µg/mL, ThermoFisher Scientific). Images were captured by epifluorescence microscopy using an Olympus IX83 inverted microscope system equipped with a UPlanSApo 60× (1.35 N.A.) oil immersion objective, pE-800 LED light source (CoolLED), appropriate fluorescent filter sets (BrightLine Pinkel DAPI/FITC/TRITC/Cy5 and CFP/YFP/mCherry multiband filter sets; Semrock) and an iXon Life 888 EMCCD camera (Andor), controlled using cellSens Dimensions (vers. 3.2) software (Olympus). Immunostained cells were also analyzed by confocal microscopy (Zeiss LSM 880 with Airyscan) with a 63× objective lens at 2× zoom, with Z-stack analysis (0.25 µM steps) and sequential image capture.

## SDS-PAGE and western blot

At 48 hpi, cells were lysed, and 20 µg of protein was subjected to reducing discontinuous sodium dodecyl sulfate (SDS)-polyacrylamide gel electrophoresis (PAGE) (12% lower gel and 4% stacking gel). Proteins were electrotransferred onto nitrocellulose membranes and probed for total Vav (rabbit anti-Vav, 59 ng/mL, 1/2000 dilution, Abcam, ab40875), Vav2 (rabbit anti-Vav2, 1 µg/mL, 1/1000 dilution, Invitrogen, SC68-03), and actin (mouse anti- actin, Clone 4, 200 ng/mL, 1/5,000 dilution, Millipore, MAB1501R). Bound antibody was detected using goat anti-rabbit and goat anti-mouse horseradish peroxidase conjugate at 1/25,000 dilution. Chemiluminescence was detected using Clarity Western ECL substrate (BioRad) with the acquisition of images on the ChemiDoc Touch imaging system (BioRad). For analysis of cell signaling pathways, membranes were probed in combination for phospho p44/42 MAPK (ERK1/2; New England Biolabs #9101S) and mouse anti-β-tubulin (Invitrogen #32–2600) or total p44/42 MAPK (ERK1/2; New England Biolabs #9102) and mouse anti-β-tubulin with co-detection with IRDye 800CW donkey anti rabbit IgG (Millenium Science 926–32213) and IRDye 680RD donkey anti mouse (Millenium Science 926-68072). Blots were scanned using an Odyssey CLx Imaging System (LI-COR Biosciences).

## Statistics

Results were analyzed using GraphPad Prism 9.3.1 and are presented as the mean ± SD or ±SEM. Statistical significance was determined using paired (for MDM) or unpaired

*t* test or two-way analysis of variance (ANOVA) with Tukeys multiple comparisons test. Results with *P*-values of <0.05 were considered statistically significant.

## RESULTS

### Changes in Vav mRNA in response to DENV are donor dependent and correlate with DENV-induced inflammatory responses in MDMs

Vav1 expression is reported to be restricted mainly to leukocytes, and monocytes/macrophages are an important target for DENV-infection and host inflammatory responses. Hence, primary MDMs were DENV infected and at 48 hpi cells fixed and immunostained with a total Vav antibody, which in MDM will largely reflect Vav1. Total Vav protein was detected in both uninfected and DENV-infected cells with no major change in protein level (Fig. 1A). Both intense and diffuse regions of protein localization were observed in both uninfected and DENV-infected cells (Fig. 1A). Vav2 was also assessed, using a specific Vav2 antibody, and similarly, there was no change in Vav2 protein by immunofluorescent staining (Fig. 1B). Analysis by confocal microscopy and Z-stacking did not demonstrate any co-localization of total Vav or Vav2 with DENV NS1 (Fig. 1C). Vav1, as well as Vav2 and 3 were next analyzed at the mRNA level. Responses of Vav mRNA

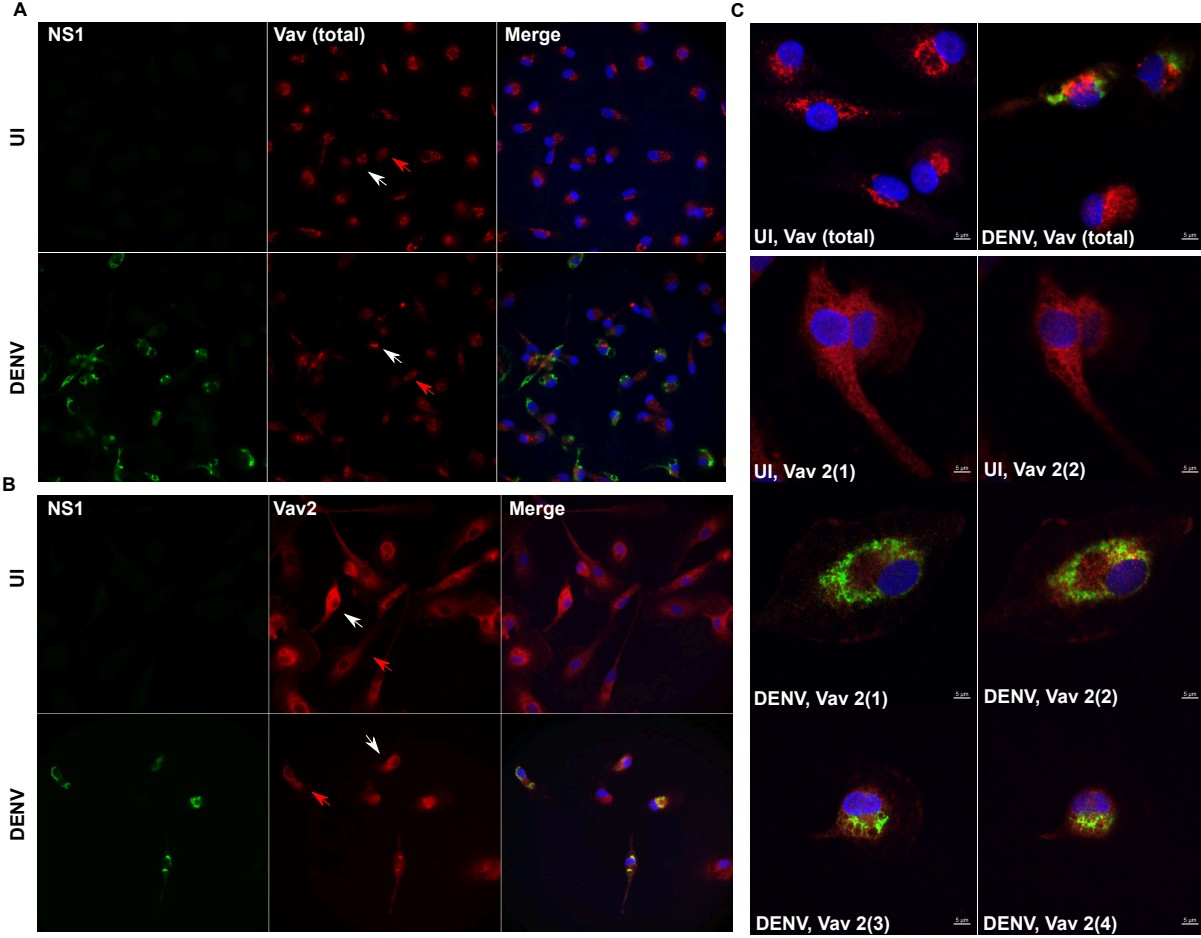

**FIG 1** Vav1 and 2 are expressed in MDMs and not affected by DENV infection. MDMs were left uninfected (UI) or DENV-infected, and at 48 hpi, cells were formalin fixed and then stained with (A) mouse-anti-NS1 (green) and rabbit-anti-Vav (total) (red) with detection of complexes using anti-mouse AlexaFluor 488 and anti-rabbit AlexaFluor 555, respectively; (B) mouse-anti-NS1 (green) and rabbit-anti-Vav2 (red) with detection of complexes using anti-mouse AlexaFluor 488 and anti-rabbit AlexaFluor 555, respectively. Nuclei were stained using Hoechst 33342 (blue). Images were captured (Olympus, IX83), and representative images are shown. (C) Images were assessed for protein colocalization by confocal microscopy (Zeiss LSM 880 with Airyscan), 63× objective lens at 2× zoom and Z-stack analysis. Two serial 0.25 µM images are shown for NS1/Vav2 staining from two different images for DENV-infected cells.

to infection were observed to differ across experiments that used different primary MDM preparations from different healthy donor blood. Representative results from two different donors that demonstrated either an increase or decrease in Vav mRNA are

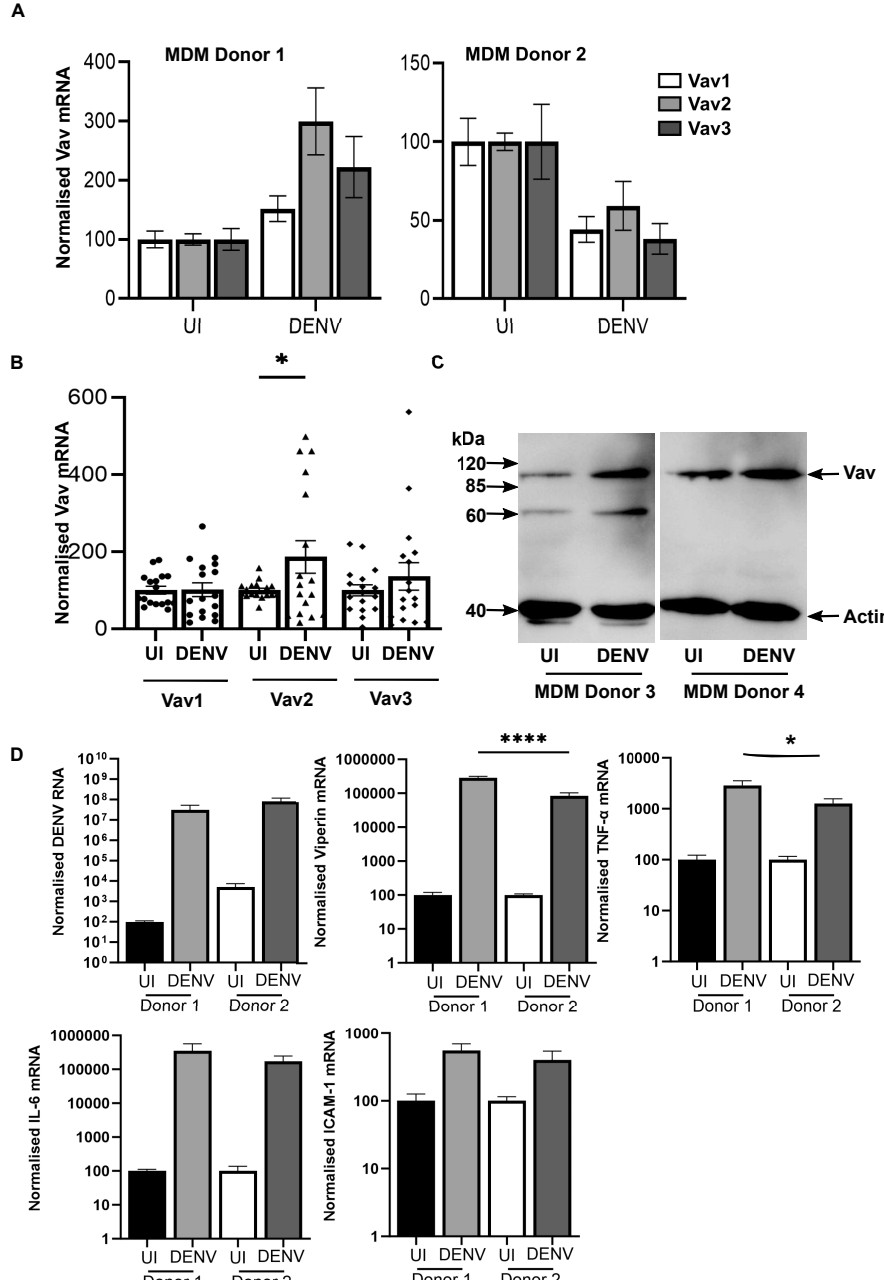

FIG 2 Vav mRNA responses to DENV-infection are dependent on the donor MDMs preparation. MDMs from different healthy donor blood were isolated, left uninfected (UI), or DENV-infected, and at 48 hpi, total RNA was extracted and subjected to qRT-PCR for Vav1, 2, and 3. Data were normalized against cyclophilin and expressed relative to measures in uninfected cells. (A) Results represent mean ± SEM from two different MDM donor preparations and infection experiments that are representative of upregulation or downregulation of Vav mRNA. (B) Results represent mean ± SEM from $n = 7$ different MDM donor preparations and infection experiments. Individual data points are shown. Data were compared by two-way ANOVA, *=$P < .05$; (C) at 48 hpi cells from two different MDM donors were lysed and 30 μg of total protein subjected to SDS-PAGE and western blot with rabbit-anti-Vav (total) and mouse-anti-actin, followed by secondary anti-rabbit-IgG or anti-mouse-IgG HRP antibody conjugates, respectively. Bound complexes were detected by chemiluminesence. (D) Analysis of DENV RNA, viperin, TNF-α, IL-6, and ICAM-1 mRNA by qRT-PCR from the two donors shown in A.

shown (Fig. 2A). The combined data from multiple donor MDMs did not show any significant difference in mRNA levels for Vav1, or 3, but Vav2 mRNA levels were increased in DENV-infected MDM (Fig. 2B). Western blot for total Vav from a further two different MDM donors also showed variation, with an apparent increase in Vav protein following DENV-infection in one donor but not a second (Fig. 2C). A smaller immunoreactive protein of approximately 60 kDa was observed in some situations (see also Fig. 4A). The level of DENV RNA and induction of antiviral (viperin), inflammatory (TNF-α, and IL-6), and cell adhesion (ICAM-1) responses in DENV-infected MDMs were quantitated by qRT-PCR. Responses in the two selected MDM donors from Fig. 2A are shown in Fig. 2D. Significantly lower induction of viperin and TNF-α was observed in donor 2 (Fig. 2D) —the donor demonstrating a decrease in Vav's following DENV infection (Fig. 2A). There was no significant difference in DENV RNA, ICAM-1, or IL-6 between these two donors (Fig. 2D). Correlation was assessed with levels of Vav1, 2, and 3 mRNA in infections across the same multiple donor MDM, as represented in Fig. 2B. Vav1, 2, or 3 mRNA levels did not correlate with DENV RNA level (Table 2). Both Vav1 and 2 positively correlated with viperin and TNF-α mRNA levels, with the correlations strongest for Vav2 (Table 2). In contrast, Vav3 mRNA levels correlated strongly with IL-6 mRNA and to a lesser extent with TNF-α and ICAM-1 (Table 2). Thus, without considering potential changes in Vav GEF activity, Vav mRNA levels differ in DENV-infected MDM from different blood donors, and this correlates with the induction of particular host responses to infection.

## Vav2 is present in ARPE-19 cells but is not influenced by DENV infection

Results above suggest an association of the level of Vav mRNA, particularly Vav2, with DENV-induced host responses, but the donor variability in MDMs makes it difficult to study this relationship further. Additionally, in primary MDM, only approximately 5%– 20% of cells are DENV-infected, and hence, there is a secondary uninfected bystander host response (Fig. 1) (36–38). THP-1 cells are a human monocyte-like cell line that can be differentiated into a macrophage-like cell type but do not support high-level DENV infection (39–41). Additionally, on-line data from the human protein atlas (https:// www.proteinatlas.org/) predict THP-1 to express Vav1, −2, and −3, which complicate a reductionist approach to study the different Vav proteins individually. ARPE-19 cells are derived from the retinal pigmented epithelium, and were chosen to assess the links between Vav and DENV-responses further, not as a model of macrophages, but as a reproducible cell line that are highly susceptible to DENV-infection with the majority of cells in the population infected, have a normal role in inflammatory responses in the eye and reflect a cell type that has relevance to DENV disease (42, 43, 44). Cells were infected and, at 48 hpi, fixed or RNA extracted. qRT-PCR outputs for Vav1 approached the limit of detection and could not be confidently quantitated, and Vav3 mRNA was not detected. Vav2 mRNA was reliably detected and not affected by DENV infection (Fig. 3A). Immunofluorescent staining of ARPE-19 cells demonstrated the presence of Vav2 in both uninfected and DENV-infected cells (Fig. 3B). Some cells were visualized with high levels of Vav2 immunostaining, but these were observed in both DENV-infected and uninfected populations (Fig. 3B). Confocal and Z-stack analysis of immunostained cells did not demonstrate any colocalization between DENV NS1 and Vav2 (Fig. 3C).

**TABLE 2** Correlation of Vav mRNA levels with inflammatory responses in DENV-infected primary MDM[a]

| | Vav1 | | | Vav2 | | | Vav3 | | |
|---|---|---|---|---|---|---|---|---|---|
| | *P*-value | *r*-value | *R*-value | *P*-value | *r*-value | *R*-value | *P*-value | *r*-value | *R*-value |
| DENV RNA | ns | | | ns | | | ns | | |
| viperin | 0.028 | 0.69 | 0.43 | 0.002 | 0.81 | 0.66 | ns | | |
| TNF-α | 0.02 | 0.69 | 0.47 | 0.0003 | 0.88 | 0.78 | 0.04 | 0.62 | 0.38 |
| IL-6 | ns | | | ns | | | 0.001 | 0.83 | 0.70 |
| ICAM-1 | ns | | | ns | | | 0.038 | 0.63 | 0.4 |

[a]For each individual donor (*n* = 7), mRNA levels were quantitated by RT-PCR, normalized to cyclophilin and correlated (Pearson's correlation). ns = not significant.

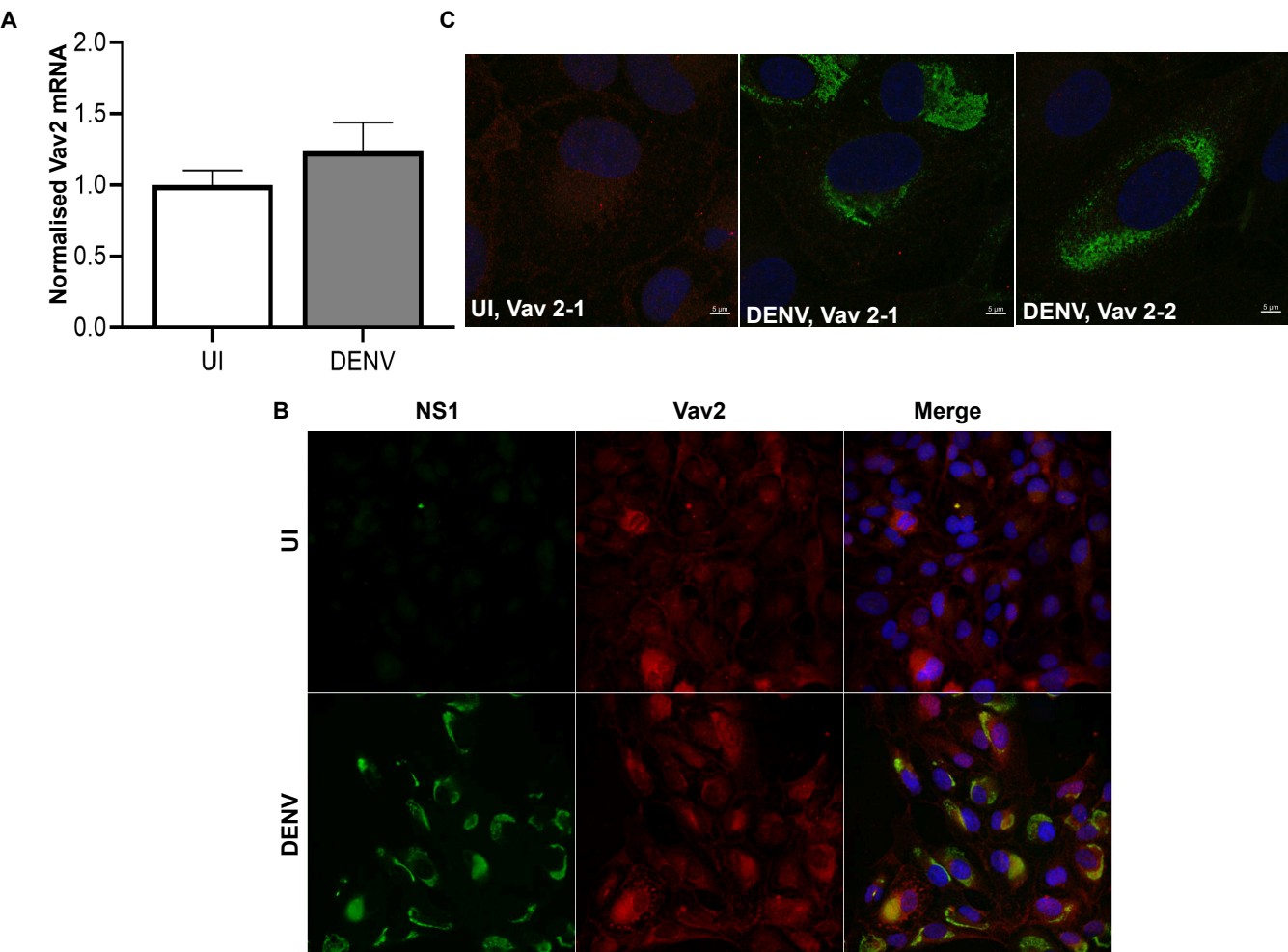

**FIG 3** Vav2 is expressed in ARPE-19 cells and not affected by DENV-infection. ARPE-19 cells were left uninfected (UI) or DENV-infected and at 48 hpi. (A) Total RNA was extracted and subjected to qRT-PCR for Vav2 mRNA. Results were normalized to cyclophilin and then expressed relative to uninfected cells. Data represent mean ± SEM from $n = 6$ measures from three independent experiments. No significant difference was observed (unpaired $t$ test with Welch's correction). (B) Cells were fixed and immunostained for mouse-anti-DENV NS1 (green) and rabbit-anti-Vav2 (red) with detection of complexes using anti-mouse 488 and anti-rabbit 555, respectively. Nuclei were stained using Hoechst 33342 (blue). Images were captured (Olympus, IX83), and representative images are shown. (C) Images were assessed for protein colocalization by confocal microscopy (Zeiss LSM 880 with Airyscan), 63X objective lens at 2X zoom and Z-stack analysis. Two different images are shown for NS1/Vav2 staining in DENV-infected cells.

ARPE19 thus represents a relatively homogeneous DENV-infected cell population, where only one kind of Vav is readily detectable offering an opportunity to manipulate endogenous Vav. Experiments were undertaken to assess if heterologous expression of Vav1 or knockdown of Vav2 in ARPE-19 cells alters DENV replication or induction of host responses. The expression of Vav1 protein was detected in transfected cells by western blot and increased Vav1 mRNA confirmed by qRT-PCR (Fig. 4A). Mock or Vav1 transfected ARPE-19 cells were DENV-infected at low MOI, RNA extracted, and subjected to qRT-PCR. Vav1 transfection did not affect DENV RNA levels or DENV-induction of viperin or IL-6 mRNA responses (Fig. 4B). TNF-α mRNA was not reliably induced by DENV-infection in ARPE-19 cells and was not assessed here. Overexpression of Vav1, however, significantly increased the levels of IL-6 mRNA in uninfected cells (Fig. 4B). siRNA knockdown of Vav-2 reduced Vav2 protein (Fig. 4C) but did not affect DENV RNA level or induction of IL-6 or viperin mRNA in DENV-infected ARPE19 cells (Fig. 4D).

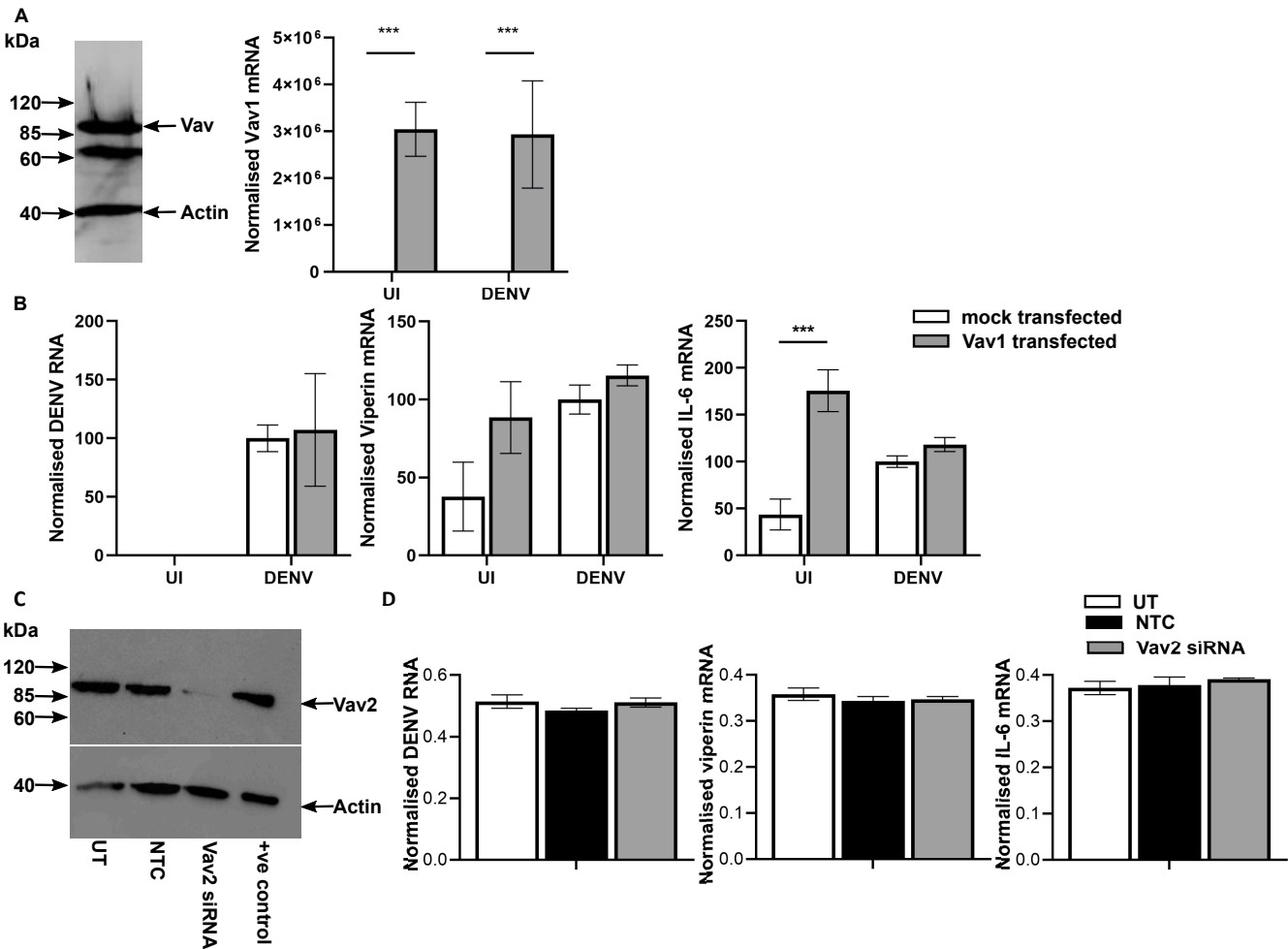

**FIG 4** Overexpression of Vav1 or siRNA knockdown of Vav2 in ARPE-19 but does not affect DENV RNA or DENV-induced responses. (A) Detection of heterologous expressed Vav1 by western blot. HA-tagged Vav1 transfected cells were lysed, and 30 µg of total protein subjected to SDS-PAGE and western blot with rabbit-anti-Vav (total) and mouse-anti-actin, followed by secondary anti-rabbit-IgG or anti-mouse-IgG HRP antibody conjugates, respectively. Bound complexes were detected by chemiluminesence. (B) ARPE-19 cells were mock or Vav1 transfected then left uninfected or DENV-infected. At 48 hpi, total RNA was extracted and subjected to qRT-PCR. Results were normalized to cyclophilin and then expressed relative to mock transfected, DENV-infected cells; qRT-PCR for DENV RNA, Vav1, viperin, and IL-6 mRNA. Data represent mean ± SEM from $n = 6$ measures from three independent experiments and were compared by two-way ANOVA, *** =$P < .005$. (C) ARPE19 cells were left untreated (UT) or treated with a non-targeted control or Vav2 siRNA. At 48 h, post treatment cells were lysed and subjected to western blot as in (A) but with rabbit-anti-Vav2. (D) At 48 hpi, total RNA was extracted and subjected to qRT-PCR. Results were normalized to cyclophilin. Data represent mean ± SEM from $n = 6$ measures from two independent experiments.

## Treatment of cells to interrupt Vav signaling does not affect DENV RNA levels but enhances DENV-mediated induction of viperin and IL-6 mRNA

The experiments above suggest that Vav's are linked to inflammation and in particular induction of IL-6. Accordingly, the impact of inhibiting Vav intracellular signaling pathways on DENV replication or induction of IL-6 and viperin was next assessed. ARPE19 were pre-treated with EHop-016, an inhibitor of the GEF activity of Vav2 for Rac1 (45), infected with DENV, cultured post-infection with EHop-016 and at 48 hpi RNA extracted for qRT-PCR. Results show a trend but no significant increase in DENV RNA or viperin mRNA but a significant increase in IL-6 mRNA levels following EHop-016 treatment (Fig. 5).

Experiments then focussed on azathioprine, a clinically available immunosuppressant. The active metabolite of azathioprine, 6-mercaptopurine, can block the interaction of p-Vav with Rac1 (46) and act as an antiviral nucleoside analog (47). Induction of pERK is

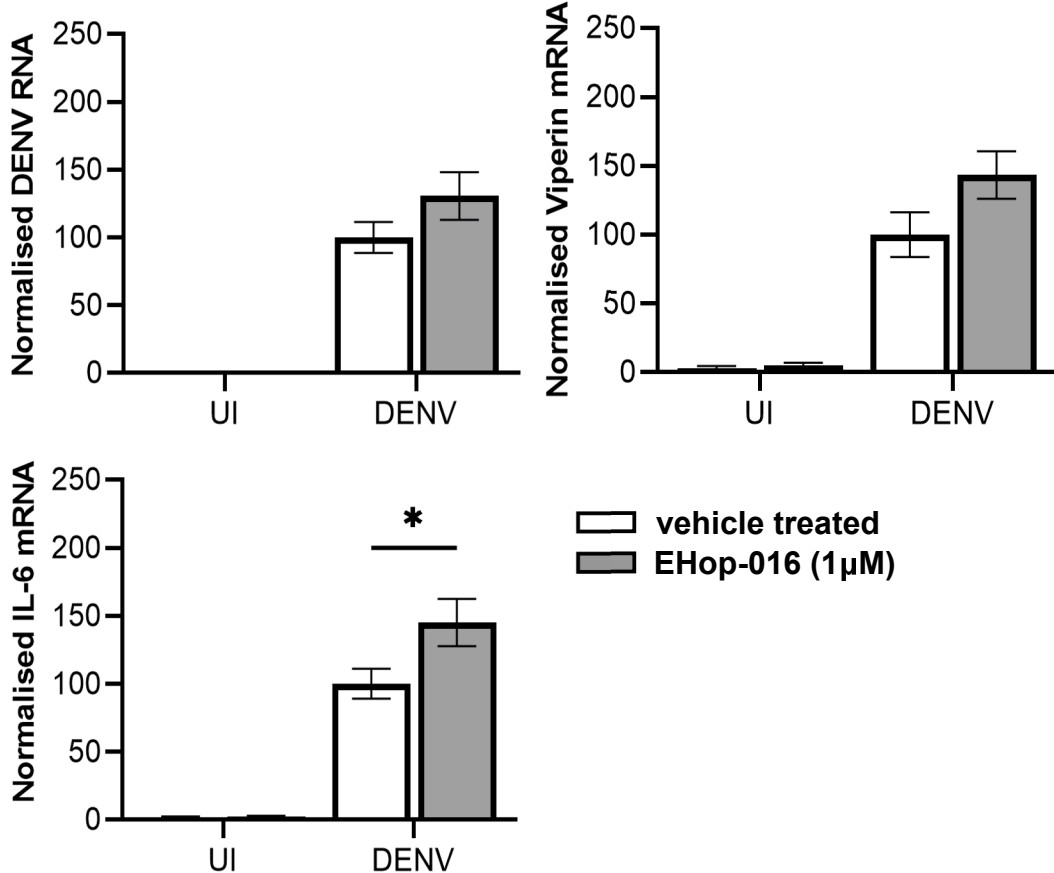

**FIG 5** EHop-016 treatment of DENV-infected ARPE-19 is not antiviral but enhances DENV-induced IL-6 mRNA. ARPE-19 cells were pre-treated with 1 μM EHop-016 or vehicle (0.01% DMSO) for 2 h then left uninfected (UI) or DENV-infected and cultured post-infection ±1 μM EHop-016. At 48 hpi, total RNA was extracted and subjected to qRT-PCR for DENV RNA or mRNA as indicated. Results were normalized to cyclophilin and then expressed relative to vehicle treated, DENV-infected cells. Data represent mean ± SEM from $n = 6$ measures from three independent experiments and were compared by two-way ANOVA, * $=P < .05$.

reduced in the absence of Vav (48) and is important for DENV infection and induction of inflammatory cytokines (49, 50), and next ARPE-19 cells were treated with azathioprine mock or DENV-infected and pERK/ERK analyzed by western blot (Fig. 6A). DENV infection induced pERK but not in DENV-infected cells treated with azathioprine (Fig. 6A and B). No change was detected in total ERK (Fig. 6A and B). Next, the effects of azathioprine on viral replication and inflammatory responses were assessed. Similar to results with EHop-016, there was no significant effect of azathioprine on DENV RNA levels but a significant increase in viperin and IL-6 mRNA levels with azathioprine treatment of DENV-infected cells (Fig. 6C).

Experiments also assessed MDMs treated with azathioprine followed by DENV infection. Consistent with the results in ARPE-19, azathioprine treatment of MDMs did not affect DENV RNA levels and enhanced DENV-mediated induction of viperin mRNA (Fig. 7). In contrast, azathioprine treatment of MDMs had no effect on DENV-induction of IL-6 (Fig. 7).

## DISCUSSION

Rho GTPase signaling pathways drive many responses that may be relevant for viruses—including inflammatory and metabolic changes (1, 3) (4). Rho GTPase signaling is regulated by GEFs, and Vav proteins are one of these GEFs that particularly influences

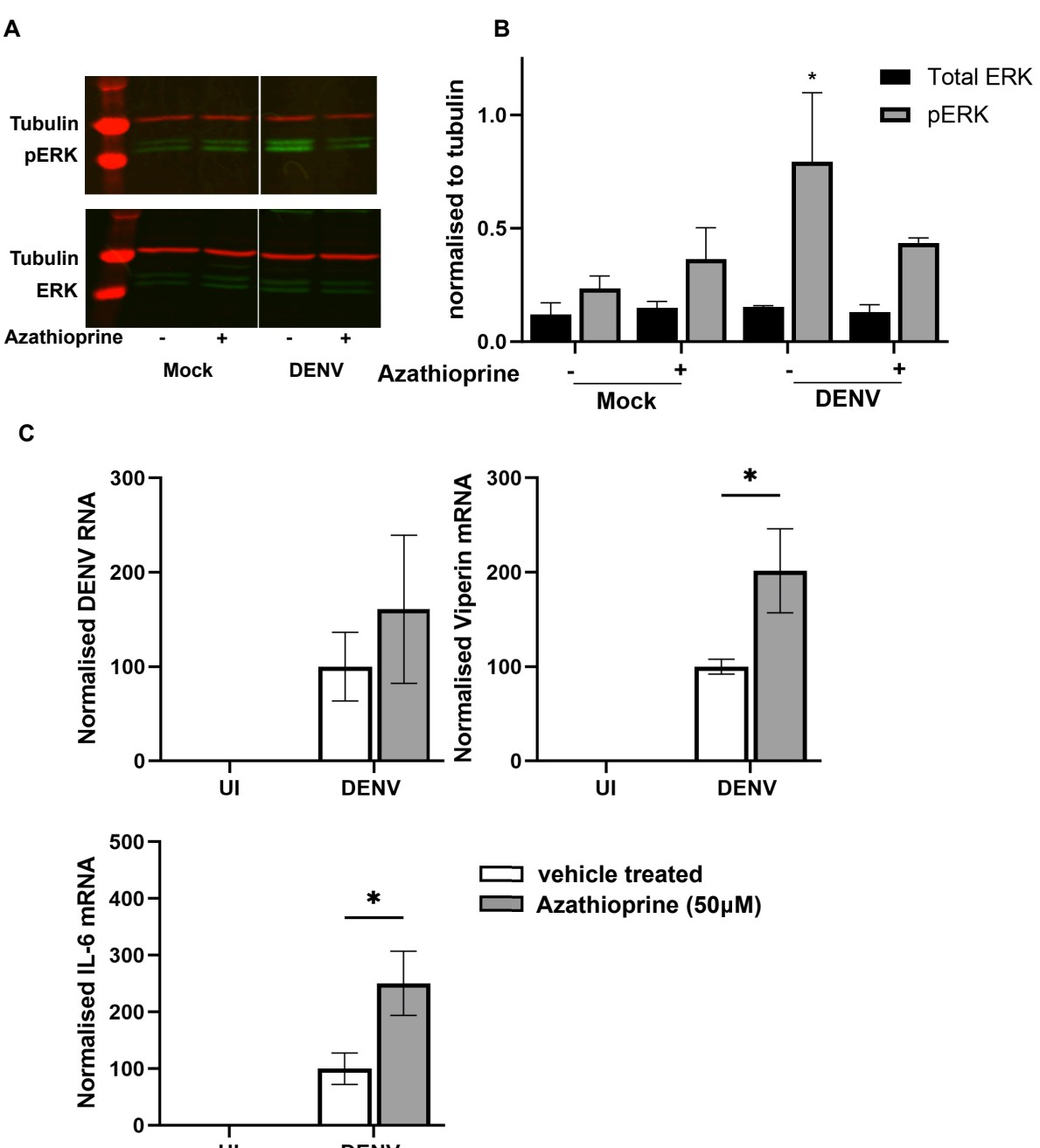

**FIG 6** Azathioprine treatment of DENV-infected ARPE-19 is not antiviral but blocks DENV-induced pERK and enhances DENV-induced viperin and IL-6 mRNA. ARPE-19 cells were pre-treated with 50 µM azathioprine or vehicle for 2 h then left uninfected (UI) or DENV-infected and cultured post-infection ±50 µM azathioprine. At 48 hpi, (A) total protein was harvested and subjected to western blot for phospho p44/42 MAPK (ERK1/2) or total p44/42 MAPK (ERK1/2; New England Biolabs #9102) in combination with mouse anti-β-tubulin. Bound complexes were co-detected with IRDye 800CW donkey anti rabbit IgG and IRDye 680RD donkey anti mouse and imaged using the Odyssey CLx Imaging System (LI-COR Biosciences). Representative images are shown. (B) Western blot images were quantitated by normalizing pERK or total ERK against tubulin. Results represent mean ± SD from $n = 2$ infections. Data were compared by two-way ANOVA, * = $P < .05$. (C) Total RNA was extracted and subjected to qRT-PCR for DENV RNA or mRNA as indicated. Results were normalized to cyclophilin and then expressed relative to vehicle treated, DENV-infected cells. Data represent mean ± SEM from $n = 6$ measures from three independent experiments and were compared by two-way ANOVA, * = $P < .05$.

Rac1, RhoA, and Cdc42 signaling (1, 5) and is important in leukocytes where Vav1 is primarily expressed (6). While the main host responses to RNA viruses such as DENV are

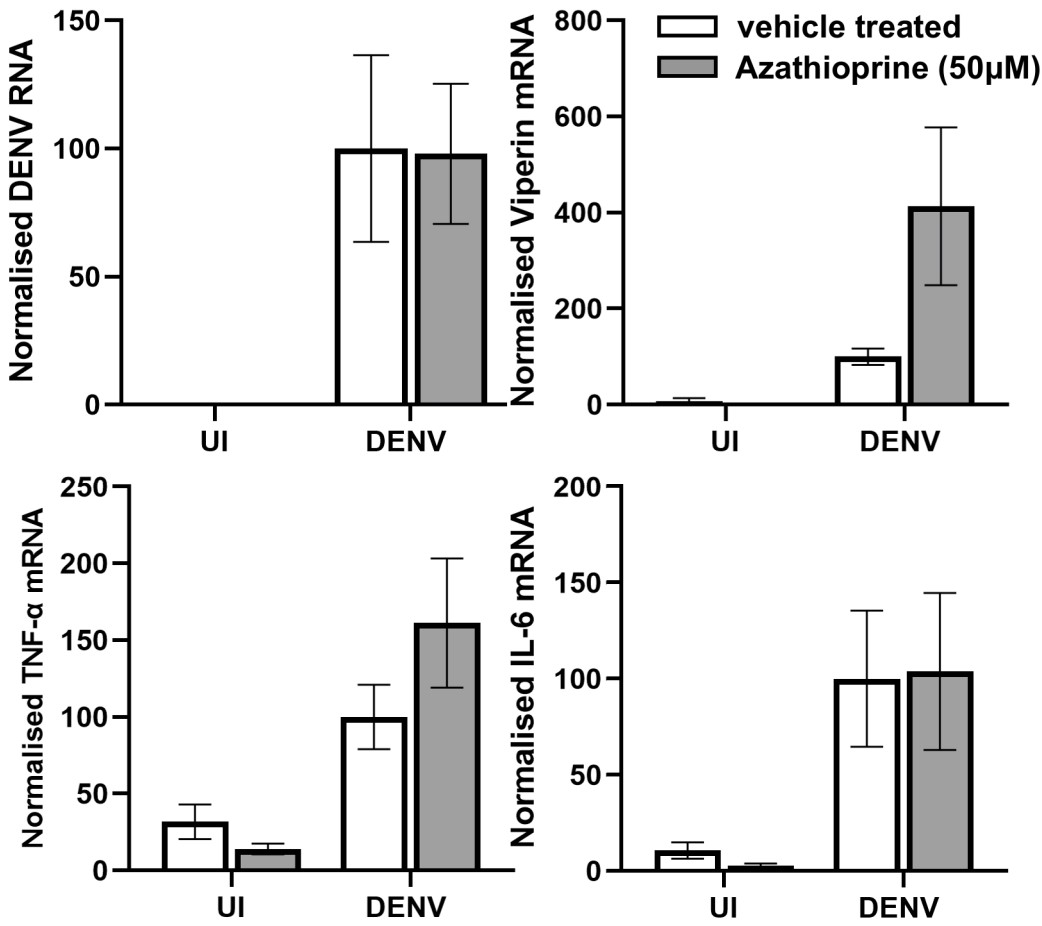

**FIG 7** Azathioprine treatment of DENV-infected MDMs is not antiviral but enhances DENV-induced viperin. MDMs were pre-treated with 50 μM azathioprine or vehicle for 2 h then left uninfected (UI) or DENV-infected and cultured post-infection ±50 μM azathioprine. At 48 hpi, total RNA was extracted and subjected to qRT-PCR for DENV RNA or mRNA as indicated. Results were normalized to cyclophilin and then expressed relative to vehicle treated, DENV-infected cells. Data represent mean ± SEM from $n = 6$ measures from three different MDM donor and infection experiments. Data were compared by two-way ANOVA, * $=P < .05$.

usually considered to be driven by activation of TLR3, RIG-I, and MDA5 (51, 52) or cGAS/STING pathways (53), other signaling pathways such as ERK/MAPK (49, 50), JNK and STAT-3 (21), and TLR4 (20) can also modulate DENV-induced responses. These latter pathways can also be affected by Vav-Rac1/RhoA signaling, and hence, the study here assessed the functional role of Vav in DENV-induced host cell responses.

Vav1 and 2 mRNA and protein and Vav3 mRNA were present in MDMs but were not affected by DENV-infection. Of note were variable mRNA changes for Vav1, 2, and 3 in different experiments from MDM preparations from different healthy blood donors. This is unlikely to be related to the degree of DENV infection in MDMs, as changes in Vav mRNA were not correlated with the level of DENV RNA. The MDMs preparations are heterogeneous and likely contain macrophages with different phenotypes and some degree of T-cell contamination (<5%) (36, 38, 54, 55), and only a low percentage of cells become DENV infected (36–38). This heterogeneity could play a role in these different Vav mRNA responses. Alternatively, these differing responses may reflect genetic polymorphisms in donor MDMs. Polymorphisms in Vav have been associated with a number of diseases such as cardiovascular disease for Vav2 and Vav3 (56), inflammatory-associated diseases such as Rheumatoid Arthritis for Vav1 (57) and C-type lectin receptor-mediated responses and susceptibility to *Candida albicans* infection with

Vav3 polymorphisms (58). Genome-wide sequence association studies have identified genetic risk as a strong predictor of dengue disease severity with multiple contributing risk alleles (59) such as *MICB* and *PLCE1* (60), BAK1 (61), MHC, and TNF-α (62). Vav polymorphisms have not previously been linked to dengue severity, and unfortunately, no ethics approval for genetic analysis of the MDM donors used here was in place to specifically test the association of Vav polymorphisms with DENV-induced inflammation in this data set. Importantly, the levels of Vav mRNA in the different donor MDMs correlated with the degree of the DENV-induced inflammatory response, with a particularly strong positive association of Vav2 with viperin and TNF-α and a positive correlation of Vav3, but not Vav1 or 2, with IL-6 and ICAM-1. Prior transcriptional profiling in the macrophage-like murine cell line, RAW 264.7, that were deficient in Vav1, 2, or 3 due to CRISPR/Cas9 knockout, showed approximately 80% conserved transcriptional profile suggesting complementing or similar roles for Vav1, 2, and 3 in the expression of the majority of genes in macrophages (63). There were, however, some interesting gene expression changes in Vav1$^{-/-}$ cells where an up regulation of Socs3 and down regulation of ISG15, CXCL10, Ifi27I2a, and OAS2 was detected suggesting an influence of Vav1 on genes known to be important in the innate immune response. Socs3 is a major inhibitor of STAT3 mediated IL-6 signaling and induction and interferon stimulated JAK-STAT signaling, and potential interactions of different Vavs with Socs3 or these signaling pathways may underpin the associations of Vavs with the ISG viperin and IL-6.

To study the Vav-DENV interaction in a simpler system, without the complication of donor variation, low percentage of infected cells, or the presence of all three Vav proteins, studies were undertaken in ARPE-19 cells representing cells from the retina. Dengue disease can be associated with inflammatory responses in the retina (42), and our laboratory has shown that primary retinal pigmented epithelial cells and ARPE-19 cells are susceptible to DENV-infection and induction of inflammatory responses (44). Hence, studies in these cells are also relevant to DENV-infection and disease. As expected from existing RNAseq data, Vav2 is a major Vav present in ARPE-19 cells (64) but was not affected at mRNA or protein level by DENV-infection. siRNA knockdown of Vav2 did not affect DENV RNA levels or DENV-induction of viperin or IL-6 mRNA. Similarly, transient transfection to introduce high-level Vav1 expression in ARPE-19 did not significantly affect DENV-mediated viperin or IL-6 mRNA induction, although overexpression of Vav1 did drive a significant increase in basal IL-6 mRNA levels. Given the above transcriptomic changes in Vav1$^{-/-}$ macrophages (63), a higher level of Vav1 would be predicted to lower levels of Socs3 and hence activate STAT3 that would promote the production of IL-6, as was observed. Additionally, Vav1 is associated with induction of IL-6 in macrophages and mast cells (65, 66). Furthermore, high Vav1 and low Socs3 would increase JAK/STAT and promote ISG induction, and in support of this, there was a non-significant trend towards increased viperin in Vav1 overexpressing ARPE-19 cells and a positive association of Vav1 with viperin in DENV-infected MDMs.

Treatment of cells was undertaken with inhibitors to block or manipulate Vav-Rac-RhoA signaling. EHop-016 was chosen since it particularly targeted the Vav2-Rac interaction. Treatment did not affect DENV RNA levels but increased DENV-induction of IL-6 mRNA. A second agent, azathioprine, was utilized, which can be metabolized into an active 6-mercaptopurine nucleoside analog that can inhibit DNA synthesis and purine biosynthesis and also bind to Rac1 and block the GDP-GTP exchange function of Vav (46). Azathioprine is used clinically as an immunomodulator in cancer therapy and inflammatory-mediated disease including in non-infectious uveitis (67, 68) and could potentially be useful for viral-induced inflammatory disease, including dengue or other viral infections of the eye. Furthermore, azathioprine's actions as a guanine antagonist could be antiviral, as reported for Zika virus (ZIKV) (47), where siRNA knockdown of RhoA and Cdc42 was shown to enhance ZIKV infection in glioblastoma cells (69). In contrast, siRNA knockdown of Rac1 and Cdc42 inhibits DENV replication in Huh-7 cells (70). Rho GTPases, including Rac1 in the context of DENV infection, have been suggested to promote viral entry via actin interactions (71, 72), and thus, there is a precedence

for manipulation of Vav-Rac-Rho signaling to impact on early viral infection. Although pre-treatment of target cells with azathioprine increased mRNA levels of the antiviral gene viperin, it did not affect DENV RNA levels in ARPE19 cells or primary MDM. Additionally, azathioprine blocked DENV induction of pERK, which would be predicted to subsequently block inflammatory cytokines (73). Here, rather than blocking inflammatory responses, azathioprine exacerbated DENV-induced IL-6 mRNA levels in ARPE-19 cells but not in primary MDM. This may be due to donor variability, as observed in Fig. 2, differences in the retinal pigmented epithelium compared to macrophage responses in particular differing host responses in uninfected bystander MDM in the DENV-challenged population, or differences due to the primary nature of MDM compared to the immortalized ARPE19 cell line. Increased IL-6 has been associated with severe dengue in many studies (74–76), and results here suggest modulating Vav does not compromise control of DENV replication but may not benefit control of IL-6 production. In addition to inflammation and antiviral responses, Vav has a role in maintaining vascular integrity and the impact of Vav on DENV-induced changes to the endothelium remains to be defined (12). Antibody-dependent enhancement (ADE) of infection, which would be expected to engage the FcϒR and potentially activate Vav/Rac/Rho signaling, results in upregulation of STAT3 and IL-6 during ZIKV and DENV infection (77) and additionally downregulates ISGs, as reviewed (78). Thus the role of Vav signaling should also be explored in the future in the context of DENV ADE.

In summary, this is the first report of the unique relationships between DENV and cellular Vav proteins. Instead of our predicted links of Vav1 with inflammation and azathioprine as an antiviral and/or anti-inflammatory agent, results have suggested potential polymorphic responses of Vav to DENV infection and roles of Vav-signaling and azathioprine in induction of pERK, IL-6, and viperin by DENV.

## ACKNOWLEDGMENTS

Thank you to Mr. Scott Shaw and Dr. Joshua Dubowsky for general assistance and to Ms. Pat Vilimas for support. The authors acknowledge the facilities and scientific and technical assistance of Microscopy Australia and the Australian National Fabrication Facility (ANFF) under the National Collaborative Research Infrastructure Strategy, at the South Australian Regional Facility, Flinders Microscopy and Microanalysis (FMMA), and Flinders University.

This study was supported by a Flinders Foundation Seed grant, and National Health and Medical Research Council (NHMRC) project grant funding (GNT1183612).

E.C., conceptual design of drug treatment studies, undertaking experiments and supervision, data analysis, generation of figures, and manuscript editing; H.J., undertaking experiments, data analysis, generation of figures, and manuscript editing; L.P.K., conceptual design of overexpression and siRNA studies, undertaking experiments, data analysis, and manuscript editing; M.G.F. for western blot analysis; V.M.S., undertaking experiments and data analysis; A.J.N., undertaking experiments, data analysis, and manuscript review; N.E., conceptual design of siRNA studies, provision of reagents and facilities, and manuscript editing; J.M.C., conceptual design of study, funding acquisition and project management, data analysis, generation of figures, and manuscript draft and editing.

## AUTHOR AFFILIATIONS

[1]College of Medicine and Public Health and Flinders Health and Medical Research Institute, Flinders University, Bedford Park, Adelaide, South Australia, Australia
[2]Faculty of Health and Medical Sciences, University of Adelaide, Adelaide, South Australia, Australia

## AUTHOR ORCIDs

Nicholas S. Eyre ⓘ http://orcid.org/0000-0002-5424-7573
Jillian M. Carr ⓘ http://orcid.org/0000-0002-1080-1472

## FUNDING

| Funder | Grant(s) | Author(s) |
|---|---|---|
| Flinders Foundation | Seed Grant | Jillian M. Carr |
| DHAC | National Health and Medical Research Council (NHMRC) | GNT1183612 | Jillian M. Carr |

## AUTHOR CONTRIBUTIONS

Evangeline Cowell, Formal analysis, Investigation, Methodology, Writing – review and editing | Hawraa Jaber, Investigation, Methodology, Writing – review and editing | Luke P. Kris, Investigation, Methodology, Writing – review and editing | Madeleine G. Fitzgerald, Investigation, Methodology | Valeria M. Sanders, Investigation, Methodology | Aidan J. Norbury, Conceptualization, Investigation, Methodology, Writing – review and editing | Nicholas S. Eyre, Conceptualization, Resources, Supervision, Writing – review and editing | Jillian M. Carr, Conceptualization, Funding acquisition, Project administration, Resources, Supervision, Writing – original draft, Writing – review and editing

## ADDITIONAL FILES

The following material is available online.

### Open Peer Review

**PEER REVIEW HISTORY (review-history.pdf).** An accounting of the reviewer comments and feedback.

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
