## [Reviewer comments · Microbiology Spectrum]

Microbiology Spectrum

Vav proteins do not influence dengue virus replication but are associated with induction of phospho-ERK, IL-6 and viperin mRNA following DENV-infection in vitro

Evangeline Cowell, Hawraa Jaber, Luke Kris, Madeleine Fitzgerald, Valeria Sanders, Aidan Norbury, Nicholas Eyre, and Jillian Carr

Corresponding Author(s): Jillian Carr, Flinders University College of Medicine and Public Health

Review Timeline:

Submission Date:	June 7, 2023
Editorial Decision:	July 20, 2023
Revision Received:	October 2, 2023
Accepted:	November 3, 2023

Editor: Manjula Kalia

Reviewer(s): The reviewers have opted to remain anonymous.

Transaction Report:

DOI: <https://doi.org/10.1128/spectrum.02391-23>

July 20, 2023

Prof. Jillian Maree Carr
Flinders University College of Medicine and Public Health
Microbiology and Infectious Diseases
College of Medicine and Public Health
Flinders Medical Centre
Adelaide, South Australia 5042
Australia

Re: Spectrum02391-23 (Vav proteins do not influence dengue virus replication but are associated with host responses and can enhance cellular IL-6 and viperin mRNA induction during infection in vitro)

Dear Prof. Jillian Maree Carr:

Thank you for submitting your manuscript to Microbiology Spectrum. Your manuscript has been reviewed by two experts. Both of them have flagged several concerns with the study, and one reviewer has recommended rejection. However, I can consider a revised version of the manuscript if you are willing to include additional experiments and data, specifically with regards to use of an appropriate cell line, and additional signalling experiments. This would be absolutely required for further consideration of the manuscript.

Link Not Available

Sincerely,

Manjula Kalia

Journals Department
Reviewer comments:

Reviewer #1 (Comments for the Author):

The manuscript tries to link Vav proteins with dengue virus mediated immune responses. Through experiments in primary monocyte-derived macrophages and in human retinal pigment

epithelial cells, the authors show that Vav inhibition does not prevent DENV replication, but rather enhances IL-6 and viperin. However, there are several technical issues with experimental design, and the model used as detailed below. The experiments are very limited in their scope, over-interpreted and the results do not support the conclusions/discussion.

Major Points:

1. A major issue of the study is that the authors have worked on two entirely different cell lines: primary monocyte derived macrophages and ARP-19 (epithelial). While ARP-19 does have an inflammatory response, it is epithelial in nature, and the extent of virus replication and the immune/inflammatory response observed here cannot be compared to macrophages.
2. Vav proteins are GEFs and their protein levels are not indicative of their activity. Hence, it is not enough to just show mRNA/protein levels without any signaling studies.
3. The rationale of expressing Vav-1 in ARP-19 cells is not clear. Does over-expression of Vav1 lead to any enhancement in its activity, or alteration in signaling?
4. The authors can consider investigating the effect of EHop-016 treatment and genetic depletion of Vavs on virus replication and immune responses in a relevant cell type such as THP-1.
5. More experiments are required to demonstrate some level of Vav mediated signaling, and its effect on the inflammatory response.

Reviewer #2 (Comments for the Author):

Review: Vav proteins do not influence dengue virus replication but are associated with host responses and can enhance cellular IL-6 and viperin mRNA induction during infection in vitro

Cowell et al., have submitted their findings reporting Vav proteins as potential influencers of DENV-induced pathogenic inflammation, which when inhibited has no impact on viral replication.

They first showed that Vav mRNA expression following DENV-infection is donor dependent and correlates with DENV-induced inflammatory responses in MDMs. In fact, the author demonstrated in immunofluorescence microscopy that Vav is colocalized with NS1 without any changes in the cellular distribution between uninfected and DENV-infected cells. They also showed by qPCR that only Vav2 mRNA was changed following DENV infection. Moreover, qPCR analyses of different cellular factor levels correlated with the mRNA level of Vav1, 2 and 3 but not with the viral RNA level. However, results were different depending on donors. They also showed by immunofluorescence microscopy and western blot that Vav2 is present in the ARPE-19 cell line but has no impact on viral replication as seen with level of viral RNA is qPCR. Finally, the authors showed that inhibition of Vav signaling enhanced induction of viperin and IL-6 expression without affecting viral RNA levels.

Major concerns:

1. Colocalization of Vav1/Vav2 and NS1 was confirmed in epifluorescence but not in confocal. Why did the authors not perform Z-stack acquisitions to show different cell layers? It is a bit odd that even when focusing for each different staining, no colocalization is observed using confocal. How do the authors explain this?
2. Donor 1 and donor 2 show different phenotypes of Vav mRNA expression in figure 1 (decrease and increase respectively). This manuscript stands that inhibition of Vav signaling could enhance DENV-mediated induction of viperin and IL-6 without changing viral RNA levels. Since donor 1 shows less Vav mRNA than donor 2, it would be great to show the level of viperin and IL-6 of these specific donors that could then better confirm the influence of Vav on IL-6 and viperin expression during DENV-infection without using heterologous expression or knockdown by siRNA.
3. The use of cell lines to confirm effects and remove the impact of donor variability is logical, however the choice of ARPE -19 cells is not well justified as a link to MDM cells. One is monocytes and the other epithelial. This does not make the results less valuable, but rather requires needs further reinforcing/justification.

Minor concerns:

Line 172: It is indicated "Vav1" while in the figure, it is indicated Vav(total). It is a bit confusing if the staining was done on Vav1 that represents Vav(total)?

Figure 1: Yellow and white arrows are difficult to differentiate. I suggest choosing another color than yellow.

The panel B in the legend is attributed to the optical zoom while in the figure, B is attributed to the staining of Vav2. Therefore, no panel C in the figure while it is indicated in the legend.

Line 240: error with reference format, it should be (1, 3, 4) instead of (1, 3) (4).

Figure 7, why were the results confirmed with MDM treated with only azathioprine, why not EHop-016 as well, this is just a curiosity, seems odd. Additionally, have these doses of drug been shown not to be toxic?

Staff Comments:

Preparing Revision Guidelines

Please return the manuscript within 60 days; if you cannot complete the modification within this time period, please contact me. If you do not wish to modify the manuscript and prefer to submit it to another journal, please notify me of your decision immediately so that the manuscript may be formally withdrawn from consideration by Microbiology Spectrum.

Review: Vav proteins do not influence dengue virus replication but are associated with host responses and can enhance cellular IL-6 and viperin mRNA induction during infection in vitro

Cowell et al., have submitted their findings reporting Vav proteins as potential influencers of DENV-induced pathogenic inflammation, which when inhibited has no impact on viral replication.

They first showed that Vav mRNA expression following DENV-infection is donor dependent and correlates with DENV-induced inflammatory responses in MDMs. In fact, the author demonstrated in immunofluorescence microscopy that Vav is colocalized with NS1 without any changes in the cellular distribution between uninfected and DENV-infected cells. They also showed by qPCR that only Vav2 mRNA was changed following DENV infection. Moreover, qPCR analyses of different cellular factor levels correlated with the mRNA level of Vav1, 2 and 3 but not with the viral RNA level. However, results were different depending on donors. They also showed by immunofluorescence microscopy and western blot that Vav2 is present in the ARPE-19 cell line but has no impact on viral replication as seen with level of viral RNA is qPCR. Finally, the authors showed that inhibition of Vav signaling enhanced induction of viperin and IL-6 expression without affecting viral RNA levels.

Major concerns:

1. Colocalization of Vav1/Vav2 and NS1 was confirmed in epifluorescence but not in confocal. Why did the authors not perform Z-stack acquisitions to show different cell layers? It is a bit odd that even when focusing for each different staining, no colocalization is observed using confocal. How do the authors explain this?
2. Donor 1 and donor 2 show different phenotypes of Vav mRNA expression in figure 1 (decrease and increase respectively). This manuscript stands that inhibition of Vav signaling could enhance DENV-mediated induction of viperin and IL-6 without changing viral RNA levels. Since donor 1 shows less Vav mRNA than donor 2, it would be great to show the level of viperin and IL-6 of these specific donors that could then better confirm the influence of Vav on IL-6 and viperin expression during DENV-infection without using heterologous expression or knockdown by siRNA.
3. The use of cell lines to confirm effects and remove the impact of donor variability is logical, however the choice of ARPE-19 cells is not well justified as a link to MDM cells. One is monocytes and the other epithelial. This does not make the results less valuable, but rather requires needs further reinforcing/justification.

Minor concerns:

Line 172: It is indicated “Vav1” while in the figure, it is indicated Vav(total). It is a bit confusing if the staining was done on Vav1 that represents Vav(total)?

Figure 1: Yellow and white arrows are difficult to differentiate. I suggest choosing another color than yellow.

The panel B in the legend is attributed to the optical zoom while in the figure, B is attributed to the staining of Vav2. Therefore, no panel C in the figure while it is indicated in the legend.

Line 240: error with reference format, it should be (1, 3, 4) instead of (1, 3) (4).

Figure 7, why were the results confirmed with MDM treated with only azathioprine, why not EHop-016 as well, this is just a curiosity, seems odd. Additionally, have these doses of drug been shown not to be toxic?

Point by point rebuttal. Spectrum02391-23; Cowell et al., Vav proteins do not influence dengue virus replication but are associated with host responses and can enhance cellular IL-6 and viperin mRNA induction during infection *in vitro*

NOTE: amended title. Vav proteins do not influence dengue virus replication but are associated with induction of phospho-ERK, IL-6 and viperin mRNA following DENV-infection *in vitro*

From the editor: I can consider a revised version of the manuscript if you are willing to include additional experiments and data, specifically with regards to use of an appropriate cell line, and additional signalling experiments. This would be absolutely required for further consideration of the manuscript.

Response: additional data have been included in Figure 1 and 3 (confocal imaging), viperin, IL-6, TNF- α , ICAM-1 mRNA in DENV-infected MDM (Figure 2) and pERK/ERK western blot analysis (Figure 6). The use of an additional cell line is addressed in response to reviewer 1, points 1 and 4, Reviewer #2, point 3, below.

Reviewer #1:

1. A major issue of the study is that the authors have worked on two entirely different cell lines: primary monocyte derived macrophages and ARP-19 (epithelial). While ARP-19 does have an inflammatory response, it is epithelial in nature, and the extent of virus replication and the immune/inflammatory response observed here cannot be compared to macrophages.

Response: the goal of using MDM and ARPE19 was not as a comparison, but as described (lines: as 217-229), as a model of high susceptibility to DENV infection, relevance to DENV disease, and lacking expression of all 3 Vavs, so that Vav 1 and 2 could be specifically manipulated. We believe that the responses in these two cells are likely different, and more complex in DENV-MDM, where there is only a low % of infected cells and likely subsequent secondary host responses in uninfected bystander cells that may be different to those in infected cells. In the ARPE19 infection model, nearly 100% of the cells are infected at our time point for analysis.

2. Vav proteins are GEFs and their protein levels are not indicative of their activity. Hence, it is not enough to just show mRNA/protein levels without any signaling studies.

Response: as indicated (line 211-213), our results have demonstrated associations of mRNA levels. We acknowledge that GEF activity would be of interest, but these activity assays are not trivial to perform, requiring immunoprecipitation and are currently not established in our laboratory.

3. The rationale of expressing Vav-1 in ARP-19 cells is not clear. Does over-expression of Vav1 lead to any enhancement in its activity, or alteration in signaling?

Response: the rationale for expressing Vav1 in ARPE19 cells is described (line 237-239), where there is no detectable endogenous Vav1.

4. The authors can consider investigating the effect of EHop-016 treatment and genetic depletion of Vavs on virus replication and immune responses in a relevant cell type such as THP-1.

Response: We believe the primary cells and cell line utilised here are both relevant to disease and as outlined (line 217-229), are relevant to the experimental design. THP-1 cells are a monocyte-like cell line that can be differentiated *in vitro* and infected with DENV. Critical assessment of the existing literature that has utilised THP-1 cells, however indicates, poor evidence for productive replication (eg. Dapat et al., 2017, <https://doi.org/10.3390/ijms18081644>, Fig 3; Pascapurnama et al., 2017 doi: [10.3389/fmicb.2017.00521](https://doi.org/10.3389/fmicb.2017.00521)), unless cells are differentiated and polarised to M2

phenotype which then has a low level of IL-6 induction (Hwang et al., 2022, doi.org/10.1016/j.heliyon.2022.e11212). Thus, we disagree with the reviewer's suggestion that THP-1 would be a relevant cell type for our studies and feel our opinion is supported by the literature. The description for our rationale for the use of primary MDM and ARPE19 in our studies has been improved.

5. More experiments are required to demonstrate some level of Vav mediated signaling, and its effect on the inflammatory response.

Response: pERK/ERK has been analysed by western blot (Figure 6 A, B) and the rationale for this choice of signalling pathway is described (line 258-260), where pERK is not induced following simulation of Vav^{-/-} T-cells and is known to be induced in response to DENV and correlate with cytokine induction. We have included an additional author M. Fitzgerald, who undertook this analysis.

Reviewer #2

1. Colocalization of Vav1/Vav2 and NS1 was confirmed in epifluorescence but not in confocal. Why did the authors not perform Z-stack acquisitions to show different cell layers? It is a bit odd that even when focusing for each different staining, no colocalization is observed using confocal. How do the authors explain this?

Response: The epifluorescence was suggestive of colocalization but confocal analysis, which has better resolution, did not support this. Thus our conclusion is that there is no colocalization of NS1 and Vav/Vav2 and confocal Z-stack images are shown (Fig 1 and 3) to support this conclusion.

2. Donor 1 and donor 2 show different phenotypes of Vav mRNA expression in figure 1 (decrease and increase respectively). This manuscript stands that inhibition of Vav signaling could enhance DENV-mediated induction of viperin and IL-6 without changing viral RNA levels. Since donor 1 shows less Vav mRNA than donor 2, it would be great to show the level of viperin and IL-6 of these specific donors that could then better confirm the influence of Vav on IL-6 and viperin expression during DENV-infection without using heterologous expression or knockdown by siRNA.

Response: Matching TNF- α , IL-6, viperin and ICAM-1 induction for donor MDM 1 and 2 are shown (Figure 2D), highlighting significant differences in induction of TNF- α and viperin between these donors that aligns with the difference in Vav mRNA levels. Further correlation analysis for the larger cohort of donors (n=7) is shown in Table 2.

3. The use of cell lines to confirm effects and remove the impact of donor variability is logical, however the choice of ARPE -19 cells is not well justified as a link to MDM cells. One is monocytes and the other epithelial. This does not make the results less valuable, but rather requires needs further reinforcing/justification.

Response: The goal of the use of ARPE19, was not to reflect responses in MDM but, as discussed (line 215-229; 236-237), as a more simplified system for manipulating Vav 1 and 2 levels, with high level of viral replication and percentage of infected cells and without the donor variation seen in MDM.

Minor concerns:

Line 172: It is indicated "Vav1" while in the figure, it is indicated Vav(total). It is a bit confusing if the staining was done on Vav1 that represents Vav(total)?

Response: the commercial antibody is stated to detect total Vav, which in the context of MDM, Vav1 is the major contributor. This has been clarified (line 183, 188).

Figure 1: Yellow and white arrows are difficult to differentiate. I suggest choosing another color than yellow.

Response: the arrows have been removed from these figures.

The panel B in the legend is attributed to the optical zoom while in the figure, B is attributed to the staining of Vav2. Therefore, no panel C in the figure while it is indicated in the legend.

Response: the figure legend has been corrected.

Line 240: error with reference format, it should be (1, 3, 4) instead of (1, 3) (4).

Response: endnote referencing has been updated.

Figure 7, why were the results confirmed with MDM treated with only azathioprine, why not EHop-016 as well, this is just a curiosity, seems odd. Additionally, have these doses of drug been shown not to be toxic?

Response: toxicity of both EHop-016 and azathioprine were validated and the concentrations used were not cytotoxic and are consistent with concentrations used in the literature. We have tested EHop-016 on DENV-MDM, where viperin and IL-6 mRNA were not enhanced by treatment, but EHop-016 treatment remains to be tested on a larger cohort of MDM donors, which is not feasible in the time frame of this rebuttal. Our results with azathioprine treatment suggest there are likely differences in responses between MDM and ARPE19 (line 354-358) and we are reluctant to report further responses in MDM that may be misleading until the rationale for these differences are better defined.

Re: Spectrum02391-23R1 (Vav proteins do not influence dengue virus replication but are associated with induction of phospho-ERK, IL-6 and viperin mRNA following DENV-infection in vitro)

Dear Prof. Jillian Maree Carr:

Your manuscript has been accepted, and I am forwarding it to the ASM production staff for publication. Your paper will first be checked to make sure all elements meet the technical requirements. ASM staff will contact you if anything needs to be revised before copyediting and production can begin. Otherwise, you will be notified when your proofs are ready to be viewed.

Sincerely,
Manjula Kalia
Editor
Microbiology Spectrum